# Ultrastructural comparison of dendritic spine morphology preserved with cryo and chemical fixation

Hiromi Tamada[1,2,3,4], Jerome Blanc[2], Natalya Korogod[2,5], Carl CH Petersen[1]*, Graham W Knott[2]*

[1]Laboratory of Sensory Processing, Brain Mind Institute, Faculty of Life Sciences, Ecole Polytechnique Fédérale de Lausanne (EPFL), Lausanne, Switzerland; [2]Biological Electron Microscopy Facility, Faculty of Life Sciences, Ecole Polytechnique Fédérale de Lausanne (EPFL), Lausanne, Switzerland; [3]Functional Anatomy and Neuroscience, Graduate School of Medicine, Nagoya University, Nagoya, Japan; [4]Japan Society of the Promotion of Sciences (JSPS), Tokyo, Japan; [5]School of Health Sciences (HESAV), University of Applied Sciences and Arts Western Switzerland (HES-SO), Lausanne, Switzerland

**Abstract** Previously, we showed that cryo fixation of adult mouse brain tissue gave a truer representation of brain ultrastructure in comparison with a standard chemical fixation method (Korogod et al., 2015). Extracellular space matched physiological measurements, there were larger numbers of docked vesicles and less glial coverage of synapses and blood capillaries. Here, using the same preservation approaches, we compared the morphology of dendritic spines. We show that the length of the spine and the volume of its head is unchanged; however, the spine neck width is thinner by more than 30% after cryo fixation. In addition, the weak correlation between spine neck width and head volume seen after chemical fixation was not present in cryo-fixed spines. Our data suggest that spine neck geometry is independent of the spine head volume, with cryo fixation showing enhanced spine head compartmentalization and a higher predicted electrical resistance between spine head and parent dendrite.

**\*For correspondence:**
carl.petersen@epfl.ch (CCHP);
graham.knott@epfl.ch (GWK)

**Competing interests:** The authors declare that no competing interests exist.

## Introduction

In our previous paper (*Korogod et al., 2015*), we used a cryo fixation method to study the neuropil of the adult mouse cerebral cortex, comparing it with a standard chemical fixation approach that has been used for many years to preserve brain tissue for analysis with electron microscopy (e.g. *Spacek and Harris, 1997*; *Knott et al., 2002*). This study had been motivated by the work of Anton Van Harreveld (*Van Harreveld et al., 1965*; *Van Harreveld and Malhotra, 1967*; see *Korogod et al., 2015* for detailed description) who had been the first to show how fresh, cryo frozen brain tissue showed appreciable amounts of extracellular space which was not present in tissue asphyxiated for 8 min. *Korogod et al., 2015* elaborated on this result showing how cryo fixation preserved an extracellular space matching previous in vivo measurements, whereas chemically fixed tissue shows reduced extracellular space. Furthermore, synapse density and vesicle docking, as well as astrocytic processes close to synapses and around blood capillaries, were significantly different between the two fixation methods. This led us in this study to investigate the extent to which dendritic spines might be affected by chemical fixation.

The dendritic spine is a small dendritic protrusion that carries the majority of the excitatory synaptic connections in the adult brain (*Colonnier, 1968*) with a size and shape that is closely linked with

its function (reviewed by: *Holtmaat and Svoboda, 2009*; *Harris and Kater, 1994*). The larger the spine head, the larger its synapse (*Harris and Stevens, 1988*; *Arellano et al., 2007*; *Knott et al., 2006*), the number of its receptors (*Kharazia and Weinberg, 1999*; *Noguchi et al., 2005*; *Nusser et al., 1998*), and its synaptic strength (*Matsuzaki et al., 2001*). However, the synapse on the spine head is separated from the parent dendrite by the spine neck. This is often thin, compartmentalizing biochemical signals and creating a potentially significant electrical resistance between synapse and dendrite. Changes in the morphology of spine head and neck have been seen after different forms of activity including LTP induction and tetanic stimulation (*Lang et al., 2004*; *Matsuzaki et al., 2004*; *Harvey and Svoboda, 2007*; *Tønnesen et al., 2014*; *Fifková and Anderson, 1981*). The understanding of dendritic spine morphology and function has come from many years of analysis using ultrastructural and in vivo analysis with electron and light microscopy. Up until now, however, all the ultrastructural data comes from chemically fixed samples. These show that the brightness of in vivo imaged spines closely correlates with their volume measured in serial electron microscopy images (*Knott et al., 2006*; *Cane et al., 2014*; *Nägerl et al., 2007*; *El-Boustani et al., 2018*). Although the resolution of light microscopy had limited the comparison of such features of spine length, and width from living neurons, the super resolution methods of STED and FRAP have been able to make these measurements in hippocampal spines in both slices and in vivo (*Tønnesen et al., 2014*; *Pfeiffer et al., 2018*). We investigated these parameters using serial section electron microscopy to compare the sizes of spine heads, their neck lengths and neck widths between cryo and chemically fixed tissue (*Figure 1*). We show that although cryo fixation reveals unchanged spine neck length and head volume, the spine neck is significantly narrower in cryo-fixed tissue compared to chemically fixed tissue, likely having an important impact upon consideration of the communication between a spine and its parent dendrite.

## Results

Spines were analyzed from 3D models constructed from serial images taken with either TEM (cryo fixation: n = 89 spines, N = 3 mice; chemical fixation: n = 75 spines, N = 1 mouse) or FIBSEM (cryo fixation: n = 27 spines, N = 2 mice; chemical fixation: n = 75 spines, N = 2 mice). No quantitative differences were found between the results obtained by TEM and FIBSEM, and the results were therefore pooled (*Figure 1—figure supplement 1*).

The tissue was prepared with identical protocols to those used in our previous paper (*Korogod et al., 2015*). As before, these revealed very little extracellular space with chemical fixation (*Figure 1*), while cryo fixation showed elements more dispersed with significant spaces separating them. The cryo-fixed membranes also had a distinct appearance from those exposed to chemical fixative. Cryo-fixed membranes were smoothly curving and lacked wrinkles typically seen after chemical fixation. Furthermore, the cryo-fixed tissue showed many thin processes, some of which were spine necks, which form the focus of the current study (*Figure 1*).

Quantifying spine geometry, we found that the mean spine neck cross-sectional area was significantly narrower in the cryo-fixed samples (cryo fixation, $0.0159 \pm 0.0185$ $\mu m^2$; chemical fixation, $0.0284 \pm 0.0197$ $\mu m^2$; unpaired Kolmogorov-Smirnov test, p<0.0001; *Figure 2A*). The same was seen for the minimum cross-sectional area (cryo fixation, $0.01 \pm 0.016$ $\mu m^2$; chemical fixation, $0.017 \pm 0.014$ $\mu m^2$; unpaired Kolmogorov-Smirnov test, p<0.0001; *Figure 2B*). The spine neck diameter was calculated from the measurements of spine neck cross-sectional area. The mean spine neck diameter was $128.0 \pm 62.8$ nm in cryo-fixed tissue, which was significantly smaller than the mean spine neck diameter of $182.4 \pm 54.5$ nm found for chemically fixed tissue (unpaired Kolmogorov-Smirnov test, p<0.0001; *Figure 2C*), and similarly for the minimum spine neck diameter (cryo fixation, $0.07 \pm 0.057$ $\mu m$; chemical fixation, $0.107 \pm 0.037$ $\mu m$; unpaired Kolmogorov-Smirnov test, p<0.0001; *Figure 2D*). Spine length measurements showed a large diversity in each group with no statistical difference between the two fixation conditions (cryo fixation, $0.770 \pm 0.550$ $\mu m$; chemical fixation $0.836 \pm 0.569$ $\mu m$; unpaired Kolmogorov-Smirnov test, p=0.53; *Figure 2E*). Measurements of the spine head volume also showed no difference between chemically fixed and cryo-fixed tissue (cryo fixation, $0.069 \pm 0.062$ $\mu m^3$; chemical fixation, $0.081 \pm 0.087$ $\mu m^3$; unpaired Kolmogorov-Smirnov test, p=0.65; *Figure 2F*).

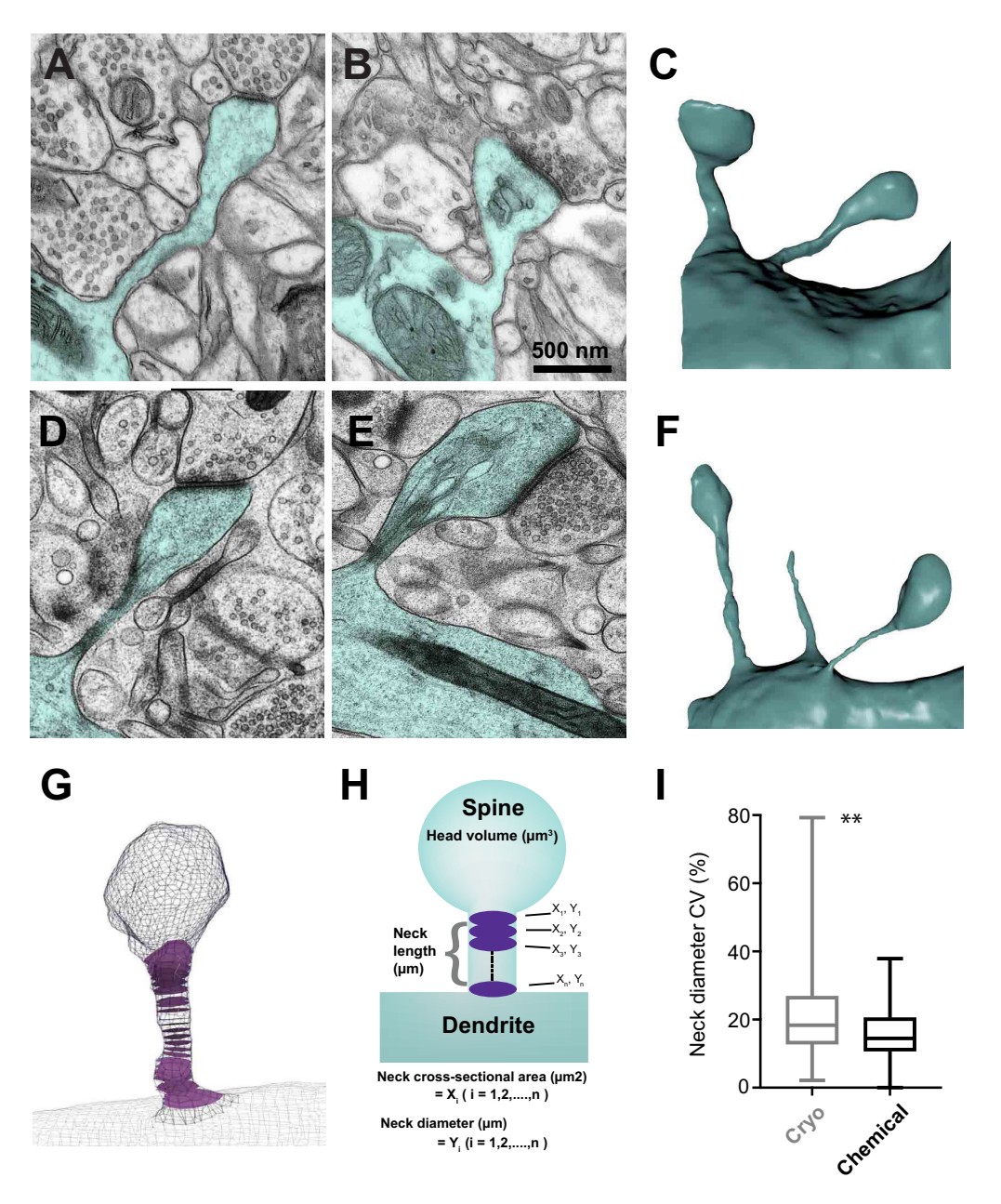

**Figure 1.** Serial section electron microscopy to compare the sizes of spine heads, their neck lengths and neck widths between cryo and chemically fixed tissue. Dendritic spines in chemically fixed (A, B, C) and cryo-fixed (D, E, F) neuropil were reconstructed in 3D from serial electron micrographs. (G) An example of a 3D model showing cross-sections (purple) through the neck region. (H) These models were used to measure head volume, neck length, and average and minimum neck cross-sectional area. The neck cross-sectional area is an average of several neck cross-sectional areas ($X_1$, $X_2$,..., $X_n$) through the spine neck, and the neck diameter ($Y_1$, $Y_2$,..., $Y_n$) was calculated from these values. (I) Box plot showing a difference in the coefficient of variation (%) of neck diameter for each spine. Plot shows the median, interquartile range with minimum and maximum values (p=0.0012, unpaired, Kolmogorov-Smirnov test).

The online version of this article includes the following source data and figure supplement(s) for figure 1:

**Source data 1.** Coefficients of variation data used in *Figure 1I*.

**Figure supplement 1.** Comparison between FIBSEM and serial section TEM data showing no significant difference between the two imaging methods.

**Figure supplement 1—source data 1.** Spine parameter data used in *Figure 1—figure supplement 1*.

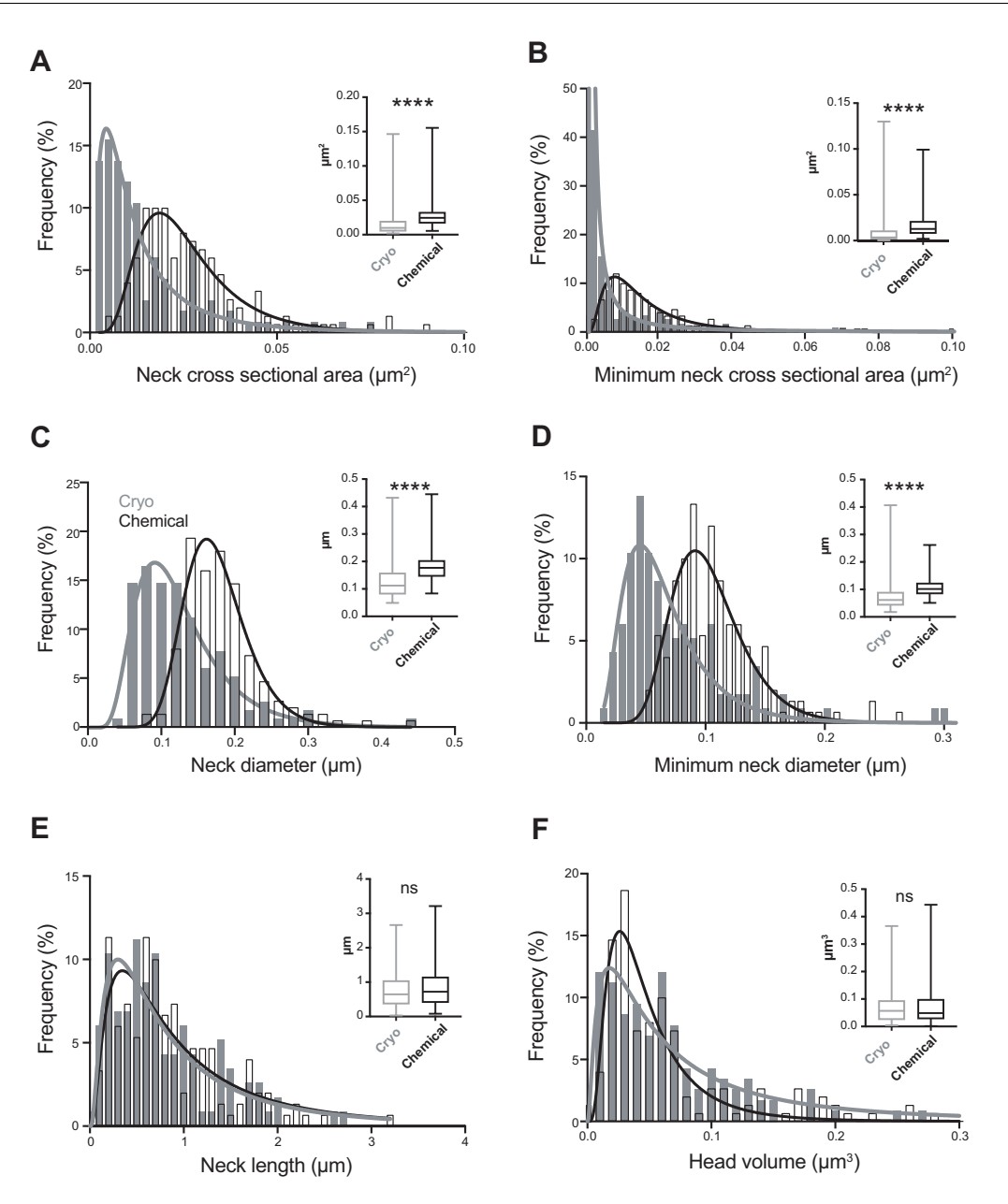

**Figure 2.** Cryo-fixed dendritic spines show reduced neck widths. (A) Distribution of neck cross-sectional area measurement in cryo-fixed (n = 116 spines, N = 5) and chemically fixed (n = 150, N = 3) cortex (p<0.0001, Kolmogorov-Smirnov). Curves show log-normal fits ($R^2$ values of 0.92 for cryo-fixed spines; 0.94 for chemically fixed). Inset shows a box plot indicating median, interquartile range and minimum and maximum values. (B) The distribution of minimum neck cross-sectional area measurements in cryo-fixed and chemically fixed cortex (p<0.0001, Kolmogorov-Smirnov). Log-normal curves have $R^2$ values of 0.98 for cryo-fixed spines and 0.98 for chemically fixed. (C) The distribution of neck diameter calculated from the cross-sectional area measurements in cryo-fixed and chemically fixed cortex (p<0.0001, Kolmogorov-Smirnov). Log-normal curves have $R^2$ values of 0.95 for cryo-fixed spines and 0.96 for chemically fixed. (D) The distribution of minimum neck diameter in cryo-fixed and chemically fixed cortex (p<0.0001, Kolmogorov-Smirnov). Log-normal curves have $R^2$ values of 0.90 for cryo-fixed spines and 0.89 for chemically fixed. (E) The distribution of neck length measurements in cryo-fixed and chemically fixed cortex (p=0.53, Kolmogorov-Smirnov). Log-normal curves have $R^2$ values of 0.80 for cryo-fixed spines and 0.76 for chemically fixed. (F) Distribution of head volume measurement in cryo-fixed and chemically fixed cortex (p=0.65, Kolmogorov-Smirnov). Log-normal curves have $R^2$ values of 0.89 for cryo-fixed spines and 0.86 for chemically fixed.

The online version of this article includes the following source data and figure supplement(s) for figure 2:

**Source data 1.** Spine parameter data used in *Figure 2*.

**Figure supplement 1.** Dendritic spine head volume correlates with the size of the synapse in cryo-fixed neuropil.

*Figure 2 continued on next page*

*Figure 2 continued*

**Figure supplement 1—source data 1.** Spine parameter data used in *Figure 2—figure supplement 1*.

Previously, we found that the synapse size, or area, was not different comparing the two fixation conditions (*Korogod et al., 2015*). Here, we further report that synapse area correlates with the volume of the spine head in cryo-fixed tissue ($R^2$ = 0.73, n = 33 spines) (*Figure 2—figure supplement 1*), similar to previous reports studying chemically fixed tissue (reviewed by *Bourne and Harris, 2008*).

We next examined correlations between spine head volume, spine neck diameter, and spine neck length (*Figure 3*). Similar to a previous study (*Arellano et al., 2007*), we found a weak correlation in the chemically fixed spines between head volume and neck diameter (r = 0.36; p<0.0001; non-parametric Spearman test). This was not observed, however, in the cryo-fixed spines (r = 0.14; p=0.12; non-parametric Spearman test). We also found no correlation between the head volume and neck length in either cryo-fixed or chemically fixed spines (cryo fixation, r = 0.124, p=0.18; chemical fixation, r = 0.044, p=0.59, non-parametric Spearman test) and also no correlation between the neck length and the neck diameter (cryo fixation, r = −0.020, p=0.83; chemical fixation, r = −0.155, p=0.059, non-parametric Spearman test).

The correlation of spine head volume and spine neck diameter in chemically fixed tissue could be due to significant structural differences in the spine neck when comparing the larger spines with smaller spines. Previous EM analyses have shown that spines with larger heads tend to be those where membranous structures, such as endoplasmic reticulum (ER), are present (*Spacek and Harris, 1997*). If, by their presence, these structures enlarge the neck then this may have an influence on the correlation between head volume and neck thickness. We investigated this question by grouping dendritic spines (chemical fixation, n = 75; cryo fix, n = 98) according to whether or not membranes were present in the neck (*Figure 4*). We scored this as ER, and did not classify them further as cisterns or vesicles as unequivocal classification in the cryo-fixed material was not possible. The cryo preparation method, where there is less extraction of molecules results in less contrast between the membranes, cytoplasm and extracellular compartments. Also, despite the sub-nanometre resolution of TEM imaging, the section thickness of 50 nm gives an obscured view of the details within the spine necks (*Figure 4A', B', C'*). FIBSEM imaging generates images with less contrast at a lower resolution. This analysis showed that spine neck length and the spine neck diameter were not different if membranes were present or absent, neither in cryo-fixed tissue nor in chemically fixed tissue (neck length, chemical fixation p=0.95, cryo fixation p=0.99; spine neck diameter, chemical fixation p=0.08, cryo fixation p=0.81; *Figure 4D,E*). Therefore, the presence of intracellular compartments within the neck do not appear to determine its thickness. However, spines in which ER is present in the spine neck do have larger heads after both cryo and chemical fixation (chemical fixation p<0.0001, cryo fixation p=0.0074; Tukey's multiple comparison test; *Figure 4F*), in agreement with a previous study of chemically fixed spines (*Spacek and Harris, 1997*).

Finally, we considered how the electrical resistance of the spines would change on the basis of the different morphological parameters measured (*Figure 5*). We used Ohm's law and the formula R = ρ × spine neck length/spine neck cross-sectional area, with a resistivity constant ρ = 109 Ω cm (*Cartailler et al., 2018*). The mean value of the resistance of spines in cryo-fixed cortex was 109.7 ± 113.4 MΩ (spine number n = 116, N = 5), and in chemically fixed cortex was 44.1 ± 37.8 MΩ (spine number n = 150, N = 3) showing a significant difference in their distributions (p<0.0001, unpaired, Kolmogorov-Smirnov test; *Figure 5A*). We found obvious correlations between resistance and spine neck cross-sectional area (cryo, r = −0.74, p<0.0001, n = 116 spines, N = 5 mice; chemical, r = −0.64, p<0.0001, n = 150 spines, N = 3 mice, correlation, non-parametric Spearman; *Figure 5B*).

## Discussion

The aim of this study was to gain a better understanding of the native ultrastructure of dendritic spines and to investigate if chemical fixation alters the morphology of dendritic spines, a neuronal structure that for many years has been the focus of considerable research due to its significant role

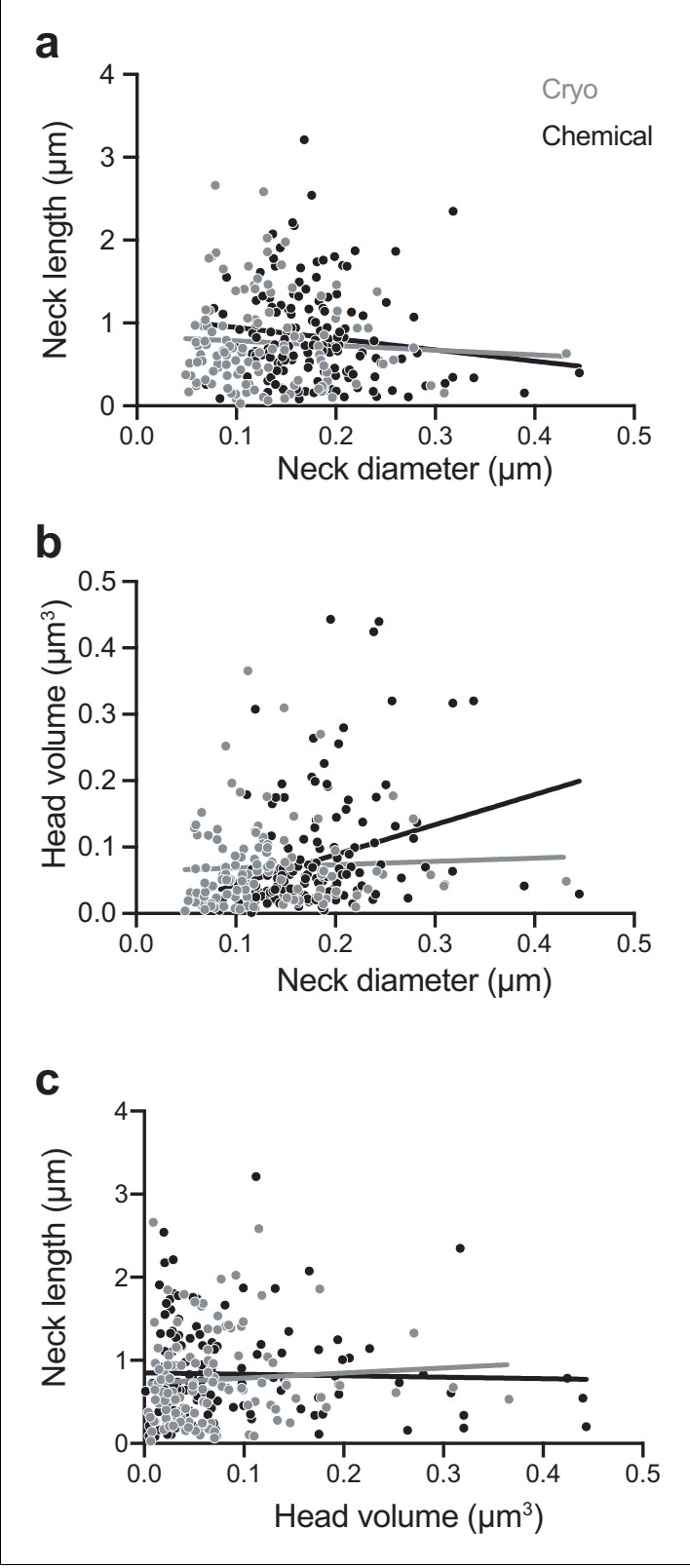

**Figure 3.** Chemically fixed, but not cryo-fixed, spines show a correlation between neck width and head volume. (A) Correlation between neck diameter and neck length for cryo-fixed spines (Spearman r = −0.020, p=0.83, n = 116 spines, N = 5 mice) and chemically fixed spines (Spearman r = −0.155, p=0.059, n = 150, N = 3 mice). (B) Correlation between neck diameter and head volume; (cryo fixed; r = 0.145, p=0.121; chemically fixed; r = 0.358,

*Figure 3 continued on next page*

*Figure 3 continued*

p<0.0001, Spearman). (C) Correlation between head volume and neck length; cryo fixed; r = 0.124, p=0.184, chemically fixed; r = 0.0441, p=0.592.

The online version of this article includes the following source data for figure 3:

**Source data 1.** Spine parameter data used in *Figure 3*.

in neuronal connectivity and plasticity (*Harris and Kater, 1994*; *Holtmaat and Svoboda, 2009*). We found that the volume of spine heads and the length of spine necks are unaltered by chemical fixation, and only the size of the spine neck is changed, being 42% thicker in diameter in chemically fixed tissue compared to cryo-fixed tissue. The thinner spine necks observed in cryo-fixed tissue suggest a larger electrical resistance coupling spines to parent dendrites than previously indicated from analysis of chemically fixed tissue.

Several of the parameters we assessed here appear to show that spines remain largely unchanged between chemical and cryo fixation, in agreement with previously published work. Spine head volume in cryo-fixed tissue was 0.069 $\mu m^3$, which was not different to the spine volume in chemically fixed tissue of 0.081 $\mu m^3$ (*Figure 2*). Our spine volume measurements are similar to other studies in the same region of the mouse brain (0.06 ± 0.04 $\mu m^3$, layer 4 [*Rodriguez-Moreno et al., 2018*]; 0.075 ± 0.005 $\mu m^3$ - 0.079 ± 0.006 $\mu m^3$, layer 1, adult and aged mice, layer 1 [*Calì et al., 2018*]). Our measurement of mean spine length after cryo-fixation of 0.77 ± 0.55 $\mu m$ was not different from the length of 0.84 ± 0.57 $\mu m$ in chemically fixed tissue (*Figure 2*), and is similar to other studies, for example rat CA1 hippocampus, 0.45 ± 0.29 $\mu m$ (*Harris and Stevens, 1989*) rat cerebellum, 0.66 ± 0.32 $\mu m$ (*Harris and Stevens, 1988*) and mouse layer 2/3 pyramidal cells, 0.66 ± 0.37 $\mu m$ (*Arellano et al., 2007*). Despite the fact that chemical fixation causes important changes to the extracellular space and astrocytic arrangement (*Korogod et al., 2015*), the overall structure of dendritic spines appears to be resilient to the alterations that the aldehydes are causing.

On the other hand, our results suggest that spine neck thickness is affected by chemical fixation, with a mean diameter of 128 nm in cryo-fixed tissue compared to 182 nm in chemically fixed tissue (*Figure 2*). This finding raises the question as to what could be happening during the aldehyde preservation method that could explain the widening. Spines are highly dynamic structures, with their shape changing during development and also adulthood (*Nimchinsky et al., 2002*), and in concert with changes in synapse size (*Sala and Segal, 2014*; *Knott et al., 2006*). To explain dynamic spine morphology, we need to consider the structural components. Actin is present throughout spines, with branched and linear actin being in different proportions in the neck and head (*Matus, 2000*; *Korobova and Svitkina, 2010*). This cytoskeleton determines the size and shape of the spine, with synaptic activation capable of altering the quantities of filamentous actin, and changing spine shape (*Honkura et al., 2008*). Modeling of dendritic spines using the physical properties of different structural components suggests the presence and arrangement of different proteins, such as actin in the neck, maintain its diameter (*Miermans et al., 2017*). This is supported by evidence of a clear periodic arrangement of actin rings in the spine neck (*Bär et al., 2016*) as well as various experiments showing that disruption of the actin regulatory pathways lead to changes in the spine shape (*Basu and Lamprecht, 2018*). However, preserving the integrity of actin is difficult, due to its sensitivity to aldehyde fixatives such as the ones used here (*Boyles et al., 1985*). Therefore, the shape of spines appears to be critically dependent on the arrangement of actin, which is dismantled on exposure to aldehydes, and this is likely to contribute to the differences in spine necks comparing cryofixed and chemically fixed tissue.

Some spine necks have also been shown to be rich in other proteins, such as the GTPases from the septin family (*Tada et al., 2007*), thought to be involved in the restriction of movement of certain membrane proteins out of the spine head (*Ewers et al., 2014*). A differential protein concentration within this narrowed part of the spine could exert on oncotic pressure pulling water into this region during chemical fixation.

We cannot exclude the possibility that chemical fixation leaves the spine neck unchanged with the differences measured occurring during the process of tissue extraction, before cryo fixation. However, we think this unlikely, as although the effects of hypoxia on dendrites have been investigated - showing changes such as swelling, the formation of filopodia, and the elongation of

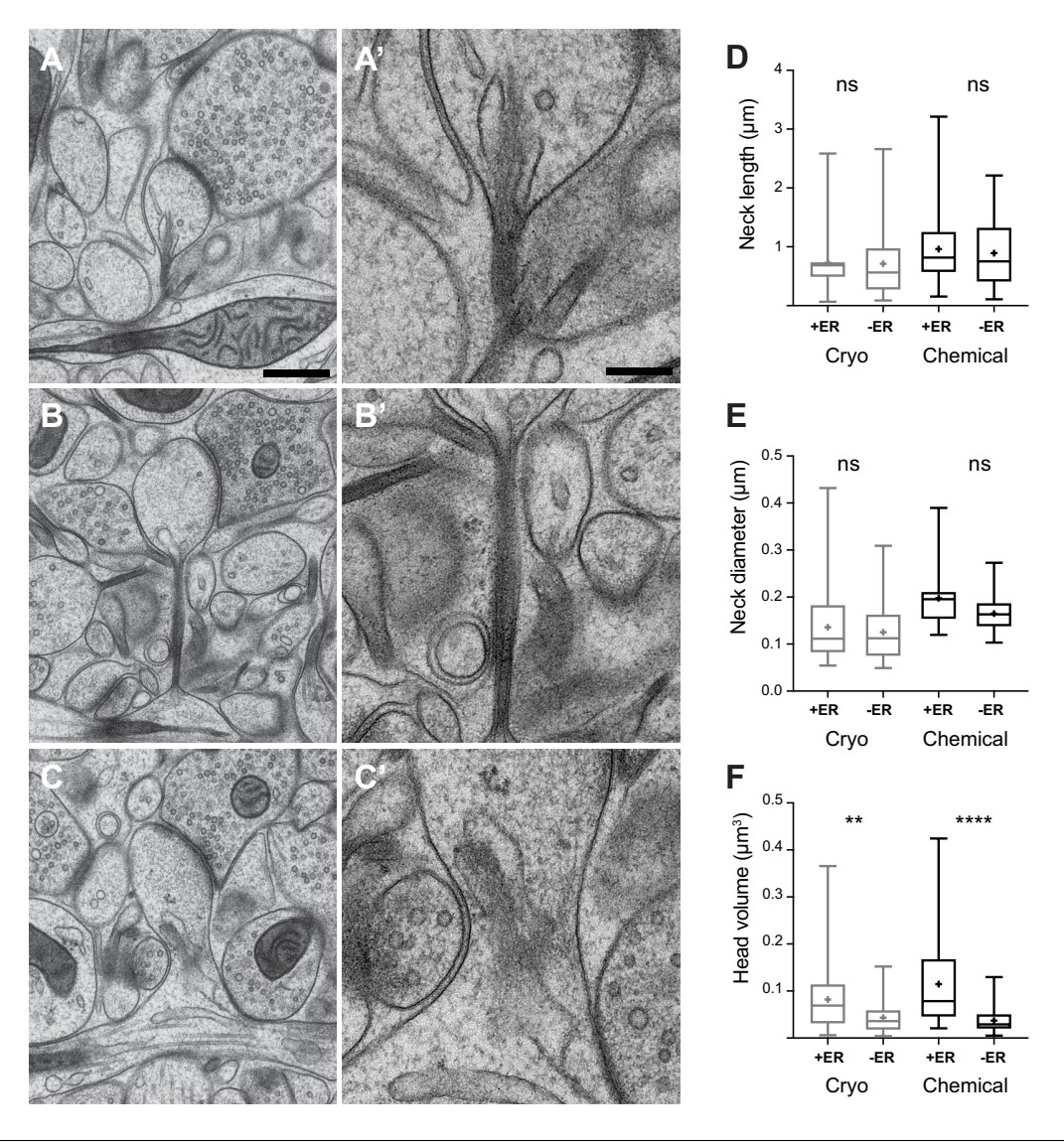

**Figure 4.** Dendritic spine geometry and the presence of membranes in the spine neck. (**A, B, C**) show examples of cryo-fixed spines, and higher magnification views of their necks (**A', B', C'**). (D) Spine neck length and, (E) spine neck diameter were not significantly different between spines with (+ER) and those without (-ER) membranes in the neck (spine neck length: cryo-fixed p=1.0, chemical fixation p=0.95; spine diameter: cryo fixation p=0.81, chemical fixation, p=0.08; Tukey's multiple comparison n = 98 cryo-fixed spines, n = 75 chemically fixed spines). (F) Spine head volume was larger in spines with membranes in the neck (cryo fixation p=0.007; chemical fixation p<0.0001; Tukey's multiple comparison). Box plots showing median, interquartile range, minimum, and maximum values, and mean (cross). Each group is divided according to whether or not membranes were present in the spine neck (+ER, or -ER). Scale bar in A is 500 nm and in A' 200 nm.

The online version of this article includes the following source data for figure 4:

**Source data 1.** Spine parameter data used in *Figure 4*.

immature spines - these were seen after 15 min (*Jourdain et al., 2002*; *Segura et al., 2016*). Our workflow vitrifies samples within 90 s of the heart stopping and within the first 30 s the tissue had been extracted and placed on a cooled surface. This method showed that in vivo levels of the extra-cellular space were preserved (*Korogod et al., 2015*) and there was no evidence of the compact neuropil, showing little extracellular space that is a feature of asphyxiated tissue as described by *Van Harreveld and Malhotra, 1967*.

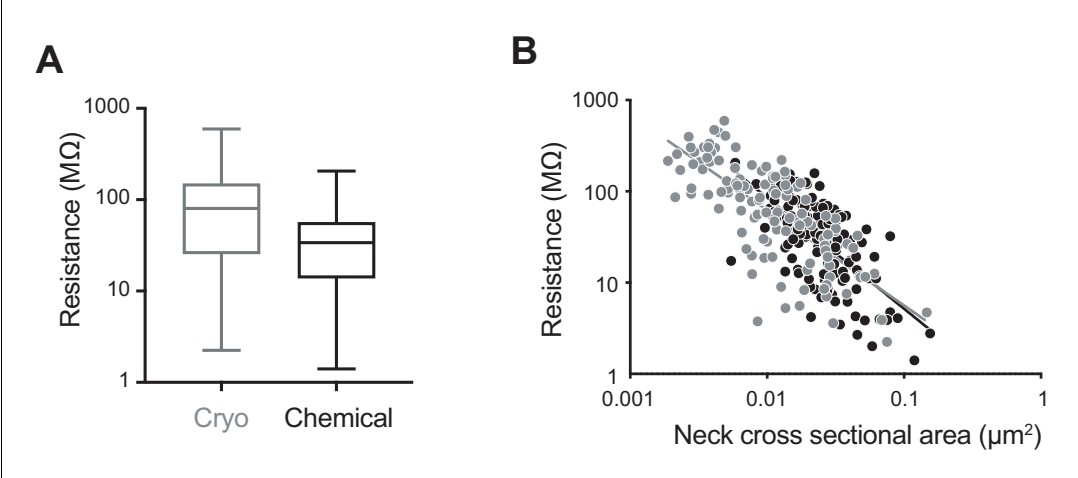

**Figure 5.** Computed spine neck resistances are higher in cryo-fixed cortex compared to chemically fixed cortex. (**A**) Dendritic spine neck resistances were calculated with resistivity constant of 109 ($\Omega$ cm) for cryo-fixed and chemically fixed cortex (cryo-fixed spines n = 116 spines, N = 5 mice; chemically fixed spines n = 150 spines, N = 3 mice). Cryo-fixed spines have higher resistance, box plots showing median, range, and distribution of resistances (p<0.0001, unpaired, Kolmogorov-Smirnov test). (**B**) Correlation between neck cross-sectional area and resistance (cryo fixed, r = −0.74, p<0.0001; chemical fixation, r = −0.64, p<0.0001).
The online version of this article includes the following source data for figure 5:

**Source data 1.** Spine parameter data used in *Figure 5*.

Our data support the observation from light microscopy showing that the morphology of the spine neck is independent of the spine head volume (*Araya et al., 2014*; *Tønnesen et al., 2014*). Previous electron microscopy analysis had suggested a correlation, although only a weak one, between these parameters (*Arellano et al., 2007*; *Bartol et al., 2015*), which we also show here with chemical fixation. However, cryo-fixed spines, presumed to be in their native state, show no correlation. Spine neck thickness, therefore, appears to be independent of the spine head volume, and therefore synapse size.

Functionally, the smaller spine widths observed in cryo-fixed tissue without change in spine neck length suggest that spine neck resistances might be higher than that implied by previous studies of chemically fixed tissue. Assuming cytosolic resistivity of 109 $\Omega$ cm (*Cartailler et al., 2018*), we estimate a mean spine neck resistance of 110 M$\Omega$ ranging from 2 M$\Omega$ to 595 M$\Omega$ in cryo-fixed tissue compared to 44 M$\Omega$ ranging from 1 M$\Omega$ to 206 M$\Omega$ in chemically fixed tissue. It is important to note that these are likely lower bound estimates, since the presence of intracellular organelles, and large macromolecular complexes, in the spine neck will further increase spine neck resistance. Our results are consistent with estimates from physiological analyses (e.g. *Bloodgood and Sabatini, 2005*; *Grunditz et al., 2008*) as well as morphological analyses in vivo using STED microscopy of hippocampal dendrites in CA1 showing neck widths of 147 nm (*Pfeiffer et al., 2018*). The same imaging approach on cultured hippocampal slices gives similar results and estimated that half of spines had a resistance of greater than 56 M$\Omega$. Here, the median value is 80 M$\Omega$ for cryo fixed, and 34 M$\Omega$ for chemically fixed spines.

At the upper end of the range of spine neck resistance, there could be a considerable impact upon spine voltage dynamics during synaptic signaling, with a 1 G$\Omega$ spine neck resistance causing a 1 mV drop in potential from spine to parent dendrite for each 1 pA of synaptic current. Given that synaptic currents are typically considered to be in the range of tens of pA, our data are consistent with tens of mV difference between spine head and dendrite for the smallest spine neck diameters. For these spines, regulation of spine neck diameter could therefore contribute to functional synaptic plasticity as previously proposed (*Bloodgood and Sabatini, 2005*; *Grunditz et al., 2008*). Our data showing smaller spine neck width in cryo-fixed tissue therefore suggest that spine neck resistance may play a more important role in synaptic signaling compared to estimates based on chemically fixed tissue.

These results, and those from our previous paper (*Korogod et al., 2015*), indicate how cryo fixation gives a truer representation of neuronal structure, at least at the level of dendritic spines. In future studies, it will be of great interest to examine axonal structure comparing cryo and chemical fixation. However, achieving a reliable preservation of neuronal tissue in its native state, and in significant volumes, is not possible with freezing due to the limited thickness to which vitrification can occur in samples with such high water content. Therefore, chemical fixation is still the best method for the preservation of significant volumes of nervous tissues, and variations to this approach that address some of its drawbacks have been used, such as alterations to osmolarity to correct for extracellular space loss (*Pallotto et al., 2015*). Our cryo fixation also uses a low-temperature embedding method with the replacement of water with solvents containing heavy metal stains. Advances in focused ion beam microscopy at cryo temperatures have shown that these steps may be avoidable in the future, removing any possible structural changes they may cause (*Schertel et al., 2013*; *Zachs et al., 2020*).

## Materials and methods

The sample preparation and imaging methods were identical to those used in our previous paper (*Korogod et al., 2015*) and described briefly below.

### Preparation of cryo-fixed tissue

This study was performed in strict accordance with the rules issued by the Swiss Federal Veterinary Office, under authorization 1889 issued by the 'Service de la consommation et des affaires vétérinaires' of the Canton de Vaud, Switzerland. The animals were handled according to approved institutional guidelines and under the experimentation license 1889.3 (Swiss Federal Veterinary Office). Adult mice (male, C57BL/6J, 6–10 weeks old, N = 5 mice) were deeply anesthetized with isoflurane (experimentation license 1889.3; Swiss Federal Veterinary Office). After decapitation, the brain was rapidly removed from the skull, placed on the top surface of a closed, cooled glass Petri dish filled with ice. The tissue was not exposed to any solutions. Razor blades were then used to slice small (approximately 2 mm x 2 mm x 0.2 mm) pieces of cortex. This was achieved by using two razor blades, one in each hand, and drawing them against each other, in opposite directions, against the surface of the glass to slice the tissues into smaller and smaller pieces. Once a piece of a suitable size is cut, it is then placed inside the 3 mm diameter and 200 µm cavity of an aluminium sample holder followed by a small drop of 1-hexadecene to remove all the air from around the piece of tissue. This was then inserted into the high-pressure freezer (HPM 100, Leica Microsystems). This entire procedure was completed in less than 90 s from the moment of decapitation.

The frozen samples, inside the aluminium carriers, were transferred to a low-temperature embedding device (ASF2; Leica Microsystems) and at −90°C the frozen pieces of tissue were transferred into a solution of 0.1% tannic acid in acetone. They were then left for 24 hr at −90°C, followed by 7 hr in 2% osmium tetroxide in acetone at −90°C. The temperature was then raised to −20°C during 14 hr, and left at this temperature for 16 hr, and then raised to 4°C during 2.4 hr and left for a further 1 hr. Next, the liquid was replaced with pure acetone and rinsed at 4°C. Finally, the tissue samples were mixed with increasing concentrations of epoxy resin (Durcupan) and left overnight at room temperature in 90% resin before 100% resin was added the next day for 6 hr, and then cured at 60°C for 24 hr.

### Preparation of chemically fixed tissue

Mice (male, C57BL/6J, 6–10 weeks old, N = 3 mice) were anesthetized with an overdose of isoflurane and transcardially perfused with 50 ml of 2.5% glutaraldehyde and 2.0% paraformaldehyde in 0.1 M phosphate buffer (pH 7.4). After 2 hr, the brain was removed and vibratome sliced at a thickness of 80 µm. Slices were then washed thoroughly with cacodylate buffer (0.1 M, pH 7.4), post-fixed for 40 min in 1.0% osmium tetroxide with 1.5% potassium ferrocyanide, and then 40 min in 1.0% osmium tetroxide alone. They were finally stained for 30 min in 1% uranyl acetate in water before being dehydrated through increasing concentrations of alcohol and then embedded in Durcupan ACM (Fluka, Switzerland) resin. The sections were then placed between glass microscope slides coated with mold releasing agent (Glorex, Switzerland) and left to harden for 24 hr in a 65°C oven.

## Electron microscopy imaging

Serial, ultrathin sections were cut at 50 nm thickness, and collected on copper grids, contrasted with lead citrate and uranyl acetate, and imaged with a transmission electron microscope (TEM) operating at 80 kV (Tecnai Spirit, FEI Company with Eagle CCD camera) with a pixel size of 2.3 nm. Additionally, focused ion beam scanning electron microscopy (FIBSEM) imaging was carried out using the same imaging parameters as described previously (*Korogod et al., 2015*). Briefly, this used an imaging voltage of 1.5 kV, and milling voltage of 30 kV and milling current of 80 pA. The pixel size was 5 nm.

The chemically fixed spines were sampled from three mice, two imaged with FIBSEM, and one with serial section TEM. The volumes imaged were 125 $\mu m^3$, 125 $\mu m^3$, and 504 $\mu m^3$, respectively. The cryo-fixed spines were sampled from five mice, two imaged with FIBSEM, and three with serial section TEM. The volumes imaged were 125 $\mu m^3$, 1112 $\mu m^3$, 504 $\mu m^3$, 450 $\mu m^3$, 446 $\mu m^3$, and 446 $\mu m^3$.

## Image processing, analysis, 3D reconstruction, and morphometric measurements

Image series from both the TEM and FIBSEM were registered using the alignment functions in the TrakEM2 plugin of FIJI (*Cardona et al., 2012*, http://fiji.sc). Spines and parts of the dendrite were segmented with the same software using the *arealist* function. All spines in each of the imaged volumes were reconstructed if their entire structure was clearly visible in all the serial images. The models were then exported as. obj files into the Blender 3D modeling software (Blender; https://www.blender.org) in which the NeuroMorph tools plugin (*Jorstad et al., 2015*; *Jorstad et al., 2018*; https://neuromorph.epfl.ch) was used to measure the spine neck cross-sectional area (minimum and average), spine neck diameter (minimum and average), spine neck length, and spine head volume (*Figure 1*).

To generate the average and minimum cross-sectional area of the spine neck, we first generated a centerline through the axis in the spine, using the software Vascular Modeling Toolkit (VMTK.org). This centerline is then used by the 'Centreline Processing' tool in the NeuroMorph addons and produces a cross-section of the spine neck at regular, defined intervals of approximately 150 nm. We defined the boundary between the spine neck and its head and the parent dendrite as the region where the spine neck expands significantly. As this measure could be considered somewhat arbitrary, we also measured the minimum spine neck thickness. The coefficient of variation of the spine neck width along the spine length was not significantly different between the two groups (cryo fixation, 21 ± 14%; chemical fixation, 15 ± 7%; unpaired Kolmogorov-Smirnov test, p=0.0012; *Figure 1I*). Therefore, there was some variation in spine neck width along their length. For this reason, both the mean and the minimum values of cross-sectional area and diameter are presented.

## Statistical analysis

All statistical analyses and plots were generated using Graph Pad Prism software. The statistical comparisons of chemically fixed vs cryo-fixed spine neck diameter, cross-section area, length and head volume were performed with unpaired, Kolmogorov-Smirnov tests. Correlation analyses between the quantified parameters were performed with non-parametric Spearman tests.

## Acknowledgements

We thank Lucie Navratilova, at the Center of Electron Microscopy EPFL, for help with the FIBSEM imaging; Catherine Maclachlan, Stéphanie Clerc-Rosset and Marie Crosier at BioEM for help with sample preparation. Funding of this work was supported by Young Researchers Exchange Programme between Japan and Switzerland (Japanese-Swiss Science and Technology Programme) (HT), JSPS KAKENHI grant #JP17KK0191 (HT), and Swiss National Science Foundation grants: #31003A_182010 (CCHP), #31003A_170082 (GWK) and CRSII3_154453 (CCHP and GWK).

## Additional information

### Funding

| Funder | Grant reference number | Author |
|---|---|---|
| Swiss National Science Foundation | 31003A_182010 | Carl CH Petersen |
| Swiss National Science Foundation | 31003A_170082 | Graham William Knott |
| Japanese Society for the Promotion of Science | JP17KK0191 | Hiromi Tamada |
| Swiss National Science Foundation | CRSII3_154453 | Carl CH Petersen Graham William Knott |

The funders had no role in study design, data collection and interpretation, or the decision to submit the work for publication.

### Author contributions

Hiromi Tamada, Data curation, Formal analysis, Investigation, Visualization, Methodology, Writing - original draft; Jerome Blanc, Formal analysis, Investigation; Natalya Korogod, Conceptualization, Methodology, Writing - review and editing; Carl CH Petersen, Conceptualization, Resources, Funding acquisition, Validation, Writing - review and editing; Graham W Knott, Conceptualization, Resources, Data curation, Supervision, Funding acquisition, Validation, Investigation, Visualization, Methodology, Writing - original draft, Project administration, Writing - review and editing

### Author ORCIDs

Carl CH Petersen https://orcid.org/0000-0003-3344-4495
Graham W Knott https://orcid.org/0000-0002-2956-9052

### Ethics

Animal experimentation: This study was performed in strict accordance with the rules issued by the Swiss Federal Veterinary Office, under authorization 1889 issued by the 'Service de la consommation et des affaires vétérinaires' of the Canton de Vaud, Switzerland. The animals were handled according to approved institutional guidelines and under the experimentation license 1889.3 (Swiss Federal Veterinary Office).

### Decision letter and Author response

Decision letter https://doi.org/10.7554/eLife.56384.sa1
Author response https://doi.org/10.7554/eLife.56384.sa2

## Additional files

### Supplementary files

• Transparent reporting form

### Data availability

All data generated during this study are included in the manuscript and the supporting files. Source data files are provided for all results. These are: Figures 1, 2, 3, 4 and 5 and Figure supplements for Figure 1 and 2.

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
