## [Decision Letter]

Thank you for submitting your article "Ultrastructural comparison of dendritic spine morphology preserved with cryo and chemical fixation" for consideration by *eLife*. Your article has been reviewed by two peer reviewers, and the evaluation has been overseen by a Reviewing Editor and Catherine Dulac as the Senior Editor. The following individual involved in review of your submission has agreed to reveal their identity: U. Valentin Nägerl (Reviewer #2).

The reviewers have discussed the reviews with one another and the Reviewing Editor has drafted this decision to help you prepare a revised submission.

This manuscript analyzes the morphology of dendritic spines in cryo-fixated brain tissue using electron microscopy (EM). Cryo fixation is known to provide a geometric arrangement of neurites that more closely resembles the in-vivo condition. In particular, it is expected to correct the reduction of extracellular space volume that the conventional aldehyde-based fixation approaches yield in EM. Interestingly, while the morphology of the spine heads (the sites of most synapses in the cerebral cortex) was largely unaltered in cryo fixated tissue, the diameter of spine necks was reduced by about 30%. Since the spine neck constitutes a biophysically relevant electrical resistance for transmission of synaptic signals towards the neuron's cell body, such a reduced diameter carries significance for arguments about spine necks as a potential regulator for effective synapse strength. Together, this work provides a focused quantitative morphological description of one of the main sites of synaptic processing in mammalian cortex.

The reviewers agreed on a few requests, all can be addressed by additional analyses or text edits:

– More careful analysis of the spine neck effect: especially comparison to thin axons to understand whether the diameter reduction is a general effect for thin neuronal processes.

– More accurate methodological descriptions, see reviews below.

– Discussion whether cryo fixation may specifically alter thin processes (i.e. could it be that cry fixation provides an overall more realistic geometric configuration, but alters thin processes disproportionally?) For this, a comparison to STED data (obtained in vivo, thus considered to be as realistic) is most relevant, see reviews below.

– It should be made clear that the comparison of cry-fixated tissue with aldehyde-based fixation is not a novel approach of the authors but was carried out by van Harreveld in the 70s. This prior art must be cited.

– It would be important to control for spine density of the parent dendritic shafts (see reviewer 1 below).

Reviewer #1:

The manuscript compared 3D EM results of spine morphology between chemical and cryo fixed samples. The most striking finding was 30% reduction in the spine neck width but with unchanged spine head volume after cryo fixation. Since the size changes appeared in a non-uniform fashion, the previously reported correction between spine neck width and spine head volume was no longer seen in the cryo fixed samples. Moreover, the modeling suggested that thinner spine neck would lead to higher electrical resistance between spine head and parental dendrite. I believe these results, when justified properly, will have considerable impact on synaptic signaling.

1) Description of individual data set was missing. How large was each data set and acquired at what resolution? How many spines were analyzed in each data set, respectively?

2) Were all spines included in the analysis or randomly sampled? If not all spines, authors need to provide more calibrations (e.g. spine density on parental dendrites). Because the traceability of thin structures was found more difficult in data set without extracellular space (Pallotto et al., 2015), sampling bias might result in a larger average spine neck width in chemical fixed samples.

3) What was the inter-section distance in Figure 1G and H? Were they equally distributed along spine neck? What was the criterion of compartmentalizing a spine? It would be also interesting to report width variation along spine neck.

4) Spine neck width was found around 150 nm using STED microscopy in acute brain slice (Tonnesen et al., 2014) and in vivo (Pfeffer et al., 2018). The authors should at least discuss this data.

5) In the subsection “Preparation of cryo-fixed tissue”: was the petri dish filled with ice or like Korogod et al., 2015, with cold ASFC? Would the pre-cooling alter the spine neck width, compared to that measured in slice and in vivo at room or physiological temperature?

6) Were the 200 μm cuts made by vibratome (as in the Korogod et al., 2015) or razor blade? Because I find 90 seconds were quite tight for a vibratome cut, considering tissue mounting, adjusting cutting windowing and knife approaching. A more detailed description of this procedure is appreciated.

Reviewer #2:

The study is an extension of the authors' 2015 e*Life* paper, where they showed that cryofixation does a much better job at preserving the extracellular space of brain tissue and astroglial morphology than standard chemical fixation. In the present study, they focus on dendritic spines of cortical neurons and compare their morphology in EM images obtained after cryogenic and chemical fixation.

The main result is that spine necks are thinner (by around one third) in cryofixed than in chemically fixed samples, most (but not all) other experimental parameters being the same. By contrast, no differences were seen for spine head size, which is reassuring.

Accurate and reliable information about the key microanatomical parameters and relationships of synapses is critical for deciphering the relationship between synaptic structure and function, and this study contributes to this effort in an important way, raising our awareness of the potential artifacts and pitfalls of the traditional approach to achieve ultrastructural perseveration for EM analyses. This is particularly relevant in a time of major efforts to reconstruct brain anatomy by EM connectomics, where people probably claim too glibly to be reporting the anatomical 'ground truth'.

The observation that spine head size is the same for both procedures indicates (but does not prove) that the measurements are valid, whereas the reported differences in spine neck diameter merely says there is a problem, but a priori we cannot tell, which procedure captures the reality more accurately, even if it seems likely that cryofixation is more trustworthy.

The paper is well written and the data are convincing. Nevertheless, it is at present a bit skinny and piecemeal. To make it more nourishing, we ask the authors to consider to also compare the shape or presence of ultrastructural features of cell-biological interest in their images, such as ER and mitochondria in spines, which may be differentially affected (relocate or remodel) in important but unknown ways during the fixation. Maybe this is possible in the spine head.

Also, it would also be very interesting to obtain analogous morphological information about axonal boutons and shafts, which may also respond differentially to the fixation procedures. The authors discuss the possibility that disruption of actin organization or oncotic pressure might account for the differences in spine neck diameters. If true, we can expect to see these effects also in other thin structures like axon shafts, which also feature actin rings, or spinules reportedly projecting out from a large fraction of spine heads. The authors should also keep the discussion more open about the 'culprit' and consider other factors (than aldehydes), e.g. potential differential ischemic conditions between the fixation procedures that may also explain the different outcomes.

---

## [Author Response]

The reviewers agreed on a few requests, all can be addressed by additional analyses or text edits:– More careful analysis of the spine neck effect: especially comparison to thin axons to understand whether the diameter reduction is a general effect for thin neuronal processes.

In the revised manuscript, we have added further careful analyses of spine neck geometry, including: minimum spine neck width and cross-sectional area, more images of cryo-fixed spines, and further details of a comparison between FIBSEM and serial section TEM imaged samples.

We agree with the reviewers that a complete and quantitative appraisal of all the elements in brain tissue should be carried out. This study, therefore, only represents the first part of this quest. We have focused initially on dendritic spines because these structures have taken centre stage for a long time in a vast literature examining the structural plasticity of the brain. Much of this work has used live imaging methods, and some chemical fixation, but without any clear understanding as to what happens to spine shape during the preservation process using aldehydes. This was the motivation of our study. We agree with the reviewers that investigating the effect of chemical fixation on axons is also of great interest, and, indeed, we are collaborating with the group of Shigeki Watanabe at John’s Hopkins on exactly this topic. This collaborative project is looking at how different axons are altered by chemical fixation and together we hope to submit a paper on this topic once all the analyses are complete. This will be presented as a separate piece of work. In the revised manuscript, we now explicitly write: “In future studies, it will be of great interest to examine axonal structure comparing cryo and chemical fixation.”

– More accurate methodological descriptions, see reviews below.

As our manuscript was submitted as a Research Advance, following on from the Korogod et al. paper (Korogod et al., 2015), we applied the journal’s guidelines and did not include every methodological detail that was previously published. However, we have elaborated on some of these to ensure that the procedures can be easily repeated. In addition, the journal Bio-Protocols, that collaborates with *eLife* to provide detailed methodologies, has accepted an elaborated write-up of this workflow, and I understand this can be linked to any final publication.

– Discussion whether cryo fixation may specifically alter thin processes (i.e. could it be that cry fixation provides an overall more realistic geometric configuration, but alters thin processes disproportionally?) For this, a comparison to STED data (obtained in vivo, thus considered to be as realistic) is most relevant, see reviews below.

The reviewers raise an interesting question. The distribution of spine neck areas after chemical fixation appears to closely reflect a right-ward shift compared to the distribution of spine neck areas in cryo-fixed tissue. There is, therefore, no indication of disproportionate effects. However, we can also not exclude such a phenomenon. We agree with the reviewer that STED imaging followed by electron microscopy would be an exciting future research project. In the revised manuscript, we now write on :

“Our results are consistent with estimates from physiological analyses (e.g. Bloodgood and Sabatini, 2005; Grunditz et al., 2008) as well as morphological analyses in vivo using STED microscopy of hippocampal dendrites in CA1 showing neck widths of 147 nm (Pfeiffer et al., 2018).”

– It should be made clear that the comparison of cry-fixated tissue with aldehyde-based fixation is not a novel approach of the authors but was carried out by van Harreveld in the 70s. This prior art must be cited.

We now include some statements and references in the first paragraph of the Introduction that cite the previous work of Anton Van Harreveld). This work was also cited in our parent paper, and, as mentioned above, in this Research Advance manuscript we did not want to repeat explanations that had already been given.

– It would be important to control for spine density of the parent dendritic shafts (see reviewer 1 below).

Detailed comments are given below.

Reviewer #1:The manuscript compared 3D EM results of spine morphology between chemical and cryo fixed samples. The most striking finding was 30% reduction in the spine neck width but with unchanged spine head volume after cryo fixation. Since the size changes appeared in a non-uniform fashion, the previously reported correction between spine neck width and spine head volume was no longer seen in the cryo fixed samples. Moreover, the modeling suggested that thinner spine neck would lead to higher electrical resistance between spine head and parental dendrite. I believe these results, when justified properly, will have considerable impact on synaptic signaling.1) Description of individual data set was missing. How large was each data set and acquired at what resolution? How many spines were analyzed in each data set, respectively?

These details are now given as follows:

“The chemically fixed spines were sampled from three mice, two imaged with FIBSEM, and one with serial section TEM. […]. The volumes imaged were 125 µm^3^, 1112 µm^3^, 504 µm^3^, 450 µm^3^, 446 µm^3^, and 446 µm^3^.”

2) Were all spines included in the analysis or randomly sampled? If not all spines, authors need to provide more calibrations (e.g. spine density on parental dendrites). Because the traceability of thin structures was found more difficult in data set without extracellular space (Pallotto et al., 2015), sampling bias might result in a larger average spine neck width in chemical fixed samples.

We included all spines that were completely contained in the volumes sampled, and which could also be clearly visualized and completely segmented. We considered this to be an unbiased approach to assessing the difference between the two fixation approaches. We accept that we could not analyse all the spines in the volumes of cryo fixed material. Some spines, and in particular those with very thin necks, were difficult to follow and reconstruct in the serial images, but not to detect. Pallotto et al., 2015, used block face scanning electron microscopy (SBEM; pixel resolution of 9.8 nm); a lower resolution imaging approach, for capturing very large volumes, but it is not our understanding, or experience, that TEM or FIBSEM gives images in which spines could not be detected. As the membrane contrast in the cryo-fixed tissue is lower than in chemically fixed, however, due to the contrasting reaction taking place at cryo temperatures and less extraction of native molecules, if any thin spines are missed, this would be more likely to occur in the cryo-fixed material.

We must also point out that the depth to which vitrification occurs is limited due to the high water content of adult brain tissue. Only a few microns are well preserved before ice crystal damage is clearly evident. Therefore, we have only small volumes in which to sample spines. The idea of reconstructing pieces of dendrite and making estimates of spine density is a good one, but was not possible in volumes with such limited thickness.

We now write:

“All spines in each of the imaged volumes were reconstructed if they their entire structure was clearly visible in all of the serial images.”

3) What was the inter-section distance in Figure 1G and H? Were they equally distributed along spine neck? What was the criterion of compartmentalizing a spine? It would be also interesting to report width variation along spine neck.

We agree that including the variation of the width would provide a better understanding of the spine neck morphology in the two groups of spines. This we have added to Figure 1 with a clearer explanation of how we define the neck (subsection “Image processing, analysis, 3D reconstruction, and morphometric measurements”). The cross-sectional areas that were calculated along the spine neck were always spaced equally, and were approximately 150 nm apart. Establishing an objective, and precise, measure of where the spine neck finishes and the spine head starts, across all spines, was not possible as the heads of some spines would not be detected, i.e. those whose necks expand gradually to form the head. We have tried this for the detection of axonal boutons in axonal reconstructions (Gala et al., 2017 *eLife*) but this only works for clearly defined boutons, such as those of excitatory axons. For this reason, we used the average neck cross sectional area, defining the neck as the region attaching the spine head to the shaft, and finishing where the neck’s width starts to expand appreciably to form the head the dendritic shaft. We do agree that this could be considered somewhat arbitrary, which would hold for all other published spine morphology studies to date. Therefore, we have now included of minimum neck width measurement. Both graphs indicated the same result that chemical fixed spines have wider necks.

4) Spine neck width was found around 150 nm using STED microscopy in acute brain slice (Tonnesen et al., 2014) and in vivo (Pfeffer et al., 2018 ). The authors should at least discuss this data.

The STED microscopy analysis of live dendritic spines is clearly important data for the assessment of spine resistances. We now cite these papers in the Discussion:

“Our results are consistent with estimates from physiological analyses (e.g. Bloodgood and Sabatini, 2005; Grunditz et al., 2008) as well as morphological analyses in vivo using STED microscopy of hippocampal dendrites in CA1 showing neck widths of 147 nm (Pfeiffer et al., 2018).”.

5) In the subsection “Preparation of cryo-fixed tissue”: was the petri dish filled with ice or like Korogod et al., 2015, with cold ASFC? Would the pre-cooling alter the spine neck width, compared to that measured in slice and in vivo at room or physiological temperature?

The use of cold ACSF in the study by Korogod et al. was only for the part of the study that estimated how the shape of the brain changed during the chemical fixation. The ACSF was used as a buffer for cutting fresh vibratome sections. It was not used for the cryo fixation of fresh brain tissue. We realise that this may have been confusing and now write:

“After decapitation, the brain was rapidly removed from the skull, placed on the top surface of a closed, cooled glass Petri dish filled with ice. The tissue was not exposed to any solutions.”

6) Were the 200 μm cuts made by vibratome (as in Korogod et al., 2015) or razor blade? Because I find 90 seconds were quite tight for a vibratome cut, considering tissue mounting, adjusting cutting windowing and knife approaching. A more detailed description of this procedure is appreciated.

This study did not use a vibratome, and the cuts were made using razor blades. The extraction of the tissue was rapid, and the cuts made quickly to ensure that the tissue was frozen as soon as possible to avoid the effects of anoxia. More details are now given in the Materials and methods section:

“Razor blades were then used to slice small (approximately 2 mm x 2 mm x 0.2 mm) pieces of cortex. This was achieved by using two razor blades, one in each hand, and drawing them against each other, in opposite directions, against the surface of the glass to slice the tissues into smaller and smaller pieces.”

Reviewer #2:The study is an extension of the authors' 2015 eLife paper, where they showed that cryofixation does a much better job at preserving the extracellular space of brain tissue and astroglial morphology than standard chemical fixation. In the present study, they focus on dendritic spines of cortical neurons and compare their morphology in EM images obtained after cryogenic and chemical fixation.The main result is that spine necks are thinner (by around one third) in cryofixed than in chemically fixed samples, most (but not all) other experimental parameters being the same. By contrast, no differences were seen for spine head size, which is reassuring.Accurate and reliable information about the key microanatomical parameters and relationships of synapses is critical for deciphering the relationship between synaptic structure and function, and this study contributes to this effort in an important way, raising our awareness of the potential artifacts and pitfalls of the traditional approach to achieve ultrastructural perseveration for EM analyses. This is particularly relevant in a time of major efforts to reconstruct brain anatomy by EM connectomics, where people probably claim too glibly to be reporting the anatomical 'ground truth'.The observation that spine head size is the same for both procedures indicates (but does not prove) that the measurements are valid, whereas the reported differences in spine neck diameter merely says there is a problem, but a priori we cannot tell, which procedure captures the reality more accurately, even if it seems likely that cryofixation is more trustworthy.The paper is well written and the data are convincing. Nevertheless, it is at present a bit skinny and piecemeal. To make it more nourishing, we ask the authors to consider to also compare the shape or presence of ultrastructural features of cell-biological interest in their images, such as ER and mitochondria in spines, which may be differentially affected (relocate or remodel) in important but unknown ways during the fixation. Maybe this is possible in the spine head.Also, it would also be very interesting to obtain analogous morphological information about axonal boutons and shafts, which may also respond differentially to the fixation procedures. The authors discuss the possibility that disruption of actin organization or oncotic pressure might account for the differences in spine neck diameters. If true, we can expect to see these effects also in other thin structures like axon shafts, which also feature actin rings, or spinules reportedly projecting out from a large fraction of spine heads. The authors should also keep the discussion more open about the 'culprit' and consider other factors (than aldehydes), e.g. potential differential ischemic conditions between the fixation procedures that may also explain the different outcomes.

The reviewer raises several excellent points. As we mentioned above, this was a follow-on study (eLife Research Advance format) from the paper by Korogod et al., 2015. We are in fact engaged in a larger study with the group of Shigeki Watanabe in the analysis of the effect of chemical fixation on the morphology of axons of different types. The reviewer’s suggestion of studying intracellular organelles is of great interest. However, there were no mitochondria within any of the spines and we were not able to detect any spinules. We study effects of the presence of ER in Figure 4, finding bigger spine head volumes in the presence of ER, but no difference in spine neck length or diameter.

We also agree that it is perhaps speculative to assume that the changes are only due to the aldehyde fixation and have now added some appropriate text into the Discussion. As our previous work had shown how the extracellular compartment was the same as physiological measurements, and this had been shown, by Van Harreveld, to reduce after 8 minutes of asphyxiation we assume the morphology seen after cryo fixation was close to native, though of course we cannot be certain. We also cite papers that show changes in immature dendrites occurring after 15 minutes of hypoxia:

“We cannot exclude the possibility that chemical fixation leaves the spine neck unchanged with the differences measured occurring during the process of tissue extraction, before cryo fixation. […] This method showed that in vivo levels of the extracellular space were preserved (Korogod et al., 2015) and there was no evidence of the compact neuropil, showing little extracellular space that is a feature of asphyxiated tissue as described by Van Harreveld and Malhotra, 1967.”